# Kinetic Model of Diclofenac Degradation Developed Using Multivariate Curve Resolution Method

**DOI:** 10.3390/molecules27227904

**Published:** 2022-11-15

**Authors:** Oxana Ye. Rodionova, Alexey L. Pomerantsev, Douglas N. Rutledge

**Affiliations:** 1Semenov Federal Research Center for Chemical Physics RAS, Kosygin 4, 119991 Moscow, Russia; 2Faculté de Pharmacie, Université Paris-Saclay, 17 Avenue des Sciences, 91400 Orsay, France; 3Muséum National d’Histoire Naturelle, 63 rue Buffon, 75005 Paris, France

**Keywords:** MCR-ALS, kinetic modeling, long-term natural aging, Diclofenac substance and intact tablets, NIR spectroscopy

## Abstract

This study presents the kinetic modeling of the natural long-term aging of the pharmaceutical substance as well as the intact tablets of Diclofenac. Datasets are collections of near-infrared spectra acquired from the intact tablets packed in plastic blisters and the spectra of the pure substance. Fresh samples and samples at different stages of degradation are analyzed. No methods of accelerated aging were applied. Multi-step application of MCR-ALS in its soft version followed by the kinetic modeling of the results helps to propose a generic degradation mechanism; which includes: a global kinetic model; approximations of the NIR spectra of the intermediate and product; rough estimates of rate constants. We study tablets in blister packs; exactly as they are presented in pharmacies; and this is important from a practical point of view.

## 1. Introduction

Diclofenac is a popular nonsteroidal anti-inflammatory drug used in more than 120 countries. It is the first phenylacetic acid derivative developed as an anti-inflammatory agent. Diclofenac sodium is a non-selective reversible and competitive inhibitor of cyclooxygenase, which is extensively used for its analgesic, antipyretic and anti-inflammatory activities. It is a potent drug applied for the treatment of active rheumatoid arthritis and osteoarthritis, ankylosing spondylitis, non-articular rheumatism, and sport injuries [1]. Topical Diclofenac can be considered a first-line treatment for osteoarthritis, especially for older patients [2]. Clinical trials suggest that it has a favorable side-effect profile and excellent patient tolerability. At the same time new contradictions [3] in prescribing Diclofenac are being investigated [4]. There are many works on the degradation of Diclofenac. However, they are devoted to cases of degradation in the environment (wastewater); see, for example [5,6]. To the best of our knowledge, studies of the aging of the drug in packaged tablets have not yet been conducted. In general, Diclofenac is a fairly stable medicine and manufacturers claim its shelf-life as being three years. However, there are cases when experts have observed a rapid degradation of this drug during routine monitoring or using complex laboratory tests. Every year since 2017, during routine testing, from one to three substandard batches of Diclofenac tablets have been identified. It goes without saying that any discrepancy in a medicinal product from the normal state or non-compliance with pharmacopeia requirements should be carefully considered and explained.

In a recently published paper [7], we presented the analysis of the natural long-term aging of both the API and the intact Diclofenac tablets produced by one manufacturer. The study was undertaken when unexpectedly rapid changes in the NIR spectra of the intact tablets were noticed in the course of routine monitoring of medicines following the workflow described in [8]. We showed that the NIR spectroscopic analysis of the intact tablets packaged in PVC blisters, supported by chemometrics, is a reliable method for the detection of even slight deviations of the medicine from its regular state. Independent components analysis helped to extract source signals from spectra of the composite object “a coated tablet sealed in PVC blister”. Further analysis of the NIR and MIR-ATR spectra of the Diclofenac substance confirmed that the aging process detected by the analysis of the intact tablets is directly related to the degradation of the API. In this paper, we continue to investigate this case and our new goal is to develop a kinetic model and propose a mechanism for this degradation of Diclofenac.

Aging is a thermodynamically controlled process when a less stable compound converts into a more stable one. This is a general fact, but in each particular case, the process occurs through a specific degradation mechanism. There are three main mechanisms possible: oxidation, hydrolysis, and photo-degradation. The last two are irrelevant in our case. Therefore, we consider oxygen to be the main agent of degradation.

Typically, the stability of pharmaceutical products or substances is studied under stress conditions to force accelerated aging [9]. Afterward, degradation paths and the mechanisms involved are examined [10,11,12]. This approach has an undeniable advantage, allowing researchers to control the process in terms of concentrations, temperature, humidity, and oxygen pressure. At the same time, it has a serious drawback, which is that one cannot answer the main question–whether artificial accelerated aging corresponds to natural aging.

The principal advantage of our study is that we use data that reflects the natural aging of Diclofenac over a rather long period, up to (or close to) its complete degradation. Another advantage is the inclusion of substance-only data, which adds new information, but at the same time produces new problems in the data interpretation.

The experimental approach is that the NIR measurements are carried out on the packaged tablets and on the substances at various stages of aging. The kinetic model of Diclofenac (*API*) degradation proposed is a sequential two-step first-order reaction:(1)$$API\overset{k_{1}}{\to }Inter\overset{k_{2}}{\to }Prod$$

We assume that it is valid both for tablets and substances. We also assume that the intermediate, *Inter*, and the product, *Prod*, are the same for the tablets and substances, while the rate constants *k*_1_ and *k*_2_ may be different. The chemometric tool we employ is Multivariate Curve Resolution–Alternating Least Squares (MCR-ALS) in its ‘soft’ version [13]. Neither “hard” nor “grey” approaches [14,15] are used, but the “soft” MCR results are subsequently modeled using kinetic equations [16].

All these circumstances should be taken into account in reasoning about what can and cannot be expected from the result of this research. We are going to present: a global kinetic model; approximations of the NIR spectra of the intermediate and the product; rough estimates of rate constants. At the same time, we will not be able to demonstrate: a detailed kinetic mechanism; identification of intermediates and products; accurate estimates of the rate constants.

## 2. Materials and Methods

### 2.1. Intact Tablets and Substances

All tablets and substances were acquired directly from the manufacturer of the tablets. New tablets packaged in the transparent PVC blister contain: 100 mg of Diclofenac sodium (*API*), 90 mg of microcrystalline cellulose (*MCC*), and 85 mg of other excipients and coating (*Rest*). In the course of aging, the API concentration decreases, and new components appear: intermediate (*Inter*) and product (*Prod*). Since the NIR measurements are conducted through the PVC blister, there is an additional component (*PVC*) that should also be taken into account [17,18]. We consider that this is a closed system in which the total sum of concentrations is constant.

Information about this system is given in Table 1. Five batches of the tablets were produced in February 2013 with an expiration date of February 2016. They were stored in blisters in closed cardboard boxes at room temperature, as prescribed by the manufacturer.

In the European Pharmacopeia [19], the substance is defined as sodium 2-[2-[(2,6-dichlorophenyl)amino]phenyl]acetate with formula C_14_H_10_Cl_2_NNaO_2_. A set of samples produced by a single manufacturer have been collected and monitored over the last ten years. We consider the Diclofenac substance as a closed system consisting of *API*, *Inter,* and *Prod*.

### 2.2. NIR Spectrometry

All spectra were acquired using an FT-NIR spectrometer (MPA, Bruker) in the diffuse-reflectance mode with a fiber probe over the 12,500–4000 cm^−1^ range, at a resolution of 2 cm^−1^. The informative range of 9000–4000 cm^−1^ is selected for data analysis. Three replicas for each object are averaged.

The tablets were measured through the transparent PVC blister. For the substance measurements, small optically transparent glass bottles with flat bottoms were used. In total, we collected 165 spectra of tablets and 36 spectra of substances. A more detailed description of the experiments can be found in [7]. All spectra were baseline-corrected before data processing.

### 2.3. MCR-ALS

MCR-ALS is an iterative method of data processing [13] that is based on model bi-linearity
**X** = **CS**^t^ + **E**.(2)

Here **X** is the (*I* × *J*) matrix that contains spectra of *I* samples recorded for *J* wavelengths. **C** is the (*I* × *N*) matrix of the component concentrations, and **S** is the (*J* × *N*) matrix of pure spectra. *N* is the number of pure components in the system. **E** is the (*I* × *J*) matrix, which contains variations not explained by the model.

The ALS procedure consists of two types of steps, the C-type step and the S-type step, which are repeated until convergence. At the C-type step, the value of **S** is set fixed as **S**_hat_, and the **C** matrix is calculated using the unconstrained least squares (LS) estimator
(3)$$C_{\mathrm{in}}=X{\left(S_{\mathrm{hat}}^{}\right)}^{+}$$

Here ^+^ means the pseudo-inverse of the matrix i.e., for any matrix **A**, **A**^+^ = **A**(**A**^t^**A**)^−1^. Afterward, matrix **C**_in_ is transformed into a matrix **C**_hat_ by incorporating any of the constraints chosen for the concentration profile.

For the S-type step, the value of **C** is set fixed at **C**_hat,_ and matrix **S** is found by applying a similar LS estimator
(4)$$S_{\mathrm{in}}=X^{t}{\left(C_{\mathrm{hat}}^{}\right)}^{+}$$

Subsequently, matrix **S**_in_ is transformed into matrix **S**_hat_ to account for any chosen spectral profile constraints.

All constraints are chosen to give a physicochemical meaning to the LS estimates **C**_in_ and **S**_in_ and, where it is possible, to resolve rotation and/or scaling ambiguity. We used non-negativity and closure for concentrations and non-negativity for spectra. The so-called ‘grey’ modeling procedure assumes that there are kinetic equations, which constrain the concentration profiles in matrix **C**. Kinetics are only taken into account in this study after the MCR-ALS step, without using the ‘grey’ approach within the course of MCR-ALS.

Prior knowledge regarding some spectral components is a special type of constraint. Consider the case when matrix **S** consists of two parts: the known spectra **S**_1_, and the unknown part **S**_2_, i.e., **S** = {**S**_1_, **S**_2_}. In this case, Equation (4) is replaced with the following equation
(5)$$S_{\mathrm{in}}=\left\{S_{1},\;{\left(X-C_{1}S_{1}^{t}\right)}^{t}{\left(C_{2}^{}\right)}^{+}\right\}$$
where **C**_1_ and **C**_2_ are the corresponding parts of the concentration matrix i.e., **C** = {**C**_1_, **C**_2_}. A very special case is one where all spectra are known. Then the MCR problem is solved in one C-step given by Equation (3).

The application of MCR-ALS in the current study has several specific features. Usually, in order to start the iteration procedure, we have to know the number of components *N* and have an initial approximation of matrix **S**. In our case, the number *N* is known. The NIR spectra of components *PVC*, *MCC,* and *API* have been reported in many papers and we also measured them. Thus, they are also known. Concentrations of *MCC*, *API*, and *Rest* are given in the tablet prescription, and their relative values are correspondingly 0.33 + 0.36 + 0.31 = 1.00. It is worth noting that the concentration of *PVC* is not a real concentration, but a weighting factor that corresponds to the contribution of the PVC layer to the entire spectrum of the tablet [17].

To assess an overall quality for the MCR modeling of the spectral matrix **X**, we calculate the relative lack of fit (LoF %) as follows
(6)$$\mathrm{LoF}=\left\|X-C_{\mathrm{hat}}S_{\mathrm{hat}}^{t}\right\|/\left\|X\right\|$$

## 3. Results

We suppose that in all steps, the main ‘players’ are the three main components, *API*, *Inter*, and *Prod*. In the course of multi-step MCR-ALS, the spectra for *Inter* and *Prod* estimated in the previous step are used as an initial approximation for the next step. In each step, we try to improve spectra estimations and calculate the corresponding concentrations of the components. At the end of each step, the kinetic model is built. The brief scheme is depicted in Figure 1.

### 3.1. Step 1. Substances

The first task is the MCR modeling of the API degradation using substance NIR data. We presume that this process can be represented by the sequential two-step, first-order reaction given in Equation (1). Therefore, the MCR model is comprised of three components that are *API* (spectrum and initial concentration known), *Inter* (both unknown), and *Prod* (both unknown). The ALS procedure converges to a value of LoF = 2.3%.

In this case, the kinetic modeling of the MCR outcomes is a challenging task, complicated by two issues. The first is that the substance samples are produced at very different times. Therefore, the quality of the substances is not as identical as in the case of tablets, which are produced by the same manufacturer. The second issue is that the values for the time intervals between production and measurement are not as reliable as for the tablet data. In fact, they are only known within a few months. Thus, when fitting the kinetic model given in Equation (1), not only are the rate constants estimated but also the time points are selected.

The estimated rate constants for the substance data are *k*_1_ = 0.023 and *k*_2_ = 0.01 (month^−1^). The results are presented in Figure 2.

### 3.2. Step 2. Tablets

This step consists of two sub-steps. At first, only the NIR spectra of the un-aged tablets are analyzed, and afterward, the estimation of the *Inter* and *Prod* spectra is performed.

#### 3.2.1. Step 2.1

There are two objectives for this modeling. The first is to estimate the spectrum *Rest* that comprises all unknown excipients and coating components, which make up 31% of the total mass of the tablet. The second goal is to find the ‘pseudo-concentration’ of the PVC blister that, in fact, is a weighting factor participating in the concentration matrix **C**. These objectives can be achieved using the MCR-ALS modeling of the initial part of the NIR data obtained from new, freshly manufactured tablets that do not show signs of degradation. Three batches with 10 tablets each are used for this modeling.

The MCR problem is solved for four components: *PVC*, *API*, *MCC*, and *Rest*, where three spectra (*PVC*, *API*, *MCC* in matrix **S**) and two concentrations (*API*, *MCC* in matrix **C**) are known. The unknown components searched for are the spectrum of *Rest* and the pseudo-concentration of *PVC*. The obvious natural constraints are that none of the concentrations change, that is, they are constants.

The results of MCR-ALS are shown in Figure 3. The left panel (a) shows the estimated spectrum of *Rest* together with the spectra of other components of the tablets. The right (b) panel shows the concentrations of the tablet components: known (*API*, *MCC*, and *Rest*) and estimated pseudo-concentration of the *PVC* blister.

The procedure converged with a good discrepancy that is characterized by a value of LoF = 1.7%. The left panel (a) of Figure 4 shows an example of the MCR fit for sample #25, which is the worst of the 30 samples. Nevertheless, it was fitted with the individual LoF = 5%. The right panel (b) shows the estimated spectrum of *Rest* together with the spectra of some common excipients that are expected to be in the tablet. From this plot, it can be concluded that there is talc in the *Rest*. Another component could be Hydroxypropyl cellulose (HPC), whose characteristic peaks are seen around 5280 and 4750 cm^−1^. Lactose monohydrate (LM) may also be seen in *Rest* due to its band near 8300 cm^−1^.

Afterward, we eliminate from the NIR spectra of the tablets, **X**_tab_, the part which does not change with time. This fraction includes the spectral data originating from the components *PVC*, *MCC*, and *Rest*.
(7)$$X_{\mathrm{clean}}=X_{\mathrm{tab}}-C_{\mathrm{fix}}S_{\mathrm{fix}}^{t}$$

Here matrices **C**_fix_ and **S**_fix_ include the known concentrations and spectra of the fixed components: *PVC*, *MCC*, and *Rest*. The cleaned spectra, **X**_clean_, correspond to the mixture of the API and the products of its degradation. Considering that the prescribed relative concentration of API in the fresh tablets is 0.36, the spectra found by Equation (7) should be further normalized by dividing by 0.36.

These spectra are shown in Figure 5 which presents five selected ‘clean’ spectra of the tablets stored for 0, 9, 18, 76, and 79 months.

The known spectrum of API is given as a reference, and it shows that the ‘clean’ spectrum at *t* = 0 (new, un-aged tablets) matches the API spectrum.

#### 3.2.2. Step 2.2

At this point, we are modeling the degradation of the tablets using the ‘clean’ data set. In the same way as for the substance, the MCR model comprises three components that are *API* (spectrum and initial concentration known), *Inter* (both unknown), and *Prod* (both unknown). To start the ALS procedure, we use the spectra *Inter* and *Prod* found in Step 1. The ALS procedure converges to a value of LoF = 7.1%, which is worse than the MCR on the spectral data of the substance alone. The concentrations obtained are well modeled by the proposed kinetics with rate constants *k*_1_ = 0.028 and *k*_2_ = 0.003 (month^−1^).

The kinetic equations were not used in the ALS step in the procedure. The curves of the kinetics shown in Figure 6b were subsequently fitted to the concentrations calculated by the MCR step.

### 3.3. Step 3. Tablets and Substances Combined

Since both the ‘clean’ tablet data and the substance data can be described by the same three components, the MCR models and the kinetic fits should be similar. The pure spectra of *Inter* and *Prod* found by MCR should also be very similar in the case of the tablets and the substances. Therefore, since there are some differences in the results obtained using the separate data sets, it is interesting to use the combined data in order to try to have better estimates of the spectra and kinetics. The MCR model for this combined dataset has a value of LoF = 7.9%.

Here we consider the case when the second stage $Inter\overset{k_{2}}{\to }Prod$ is different in the tablets and substances, so the constant *k*_1_ is the same, but *k*_2_ is different. The spectra and kinetic model under such a hypothesis are shown in Figure 7.

The found constants are summarized in Table 2.

From Figure 7, we can see that the first stage of the reaction is almost over at 79 months and API is close to complete degradation. As to the second stage, it is far from being completed. Thus, in the ‘clean’ spectra for 76 and 79 months (Figure 5), we observe a mixture of intermediate and final products.

## 4. Discussion and Conclusions

Looking at the experimental setup that we are using, one could argue that it should be the last in the list of approaches suitable for kinetic analysis. In fact, we were working with the following object: the mixture of the API with known components pressed into tablets, coated with a complex cover, and packaged in the PVC film. This object was placed in an NIR spectrometer. Of course, this setup gives rise to many problems in data analysis, but it monitors the natural aging of real products, and this is the only way to understand the mechanism of aging. As was said in the introduction, we consider oxygen to be the main agent of degradation. There are two aspects we have to take into account.

First, Diclofenac itself is quite resistant to oxidation. The key role in oxidation is due to its contaminants, and impurities, acting as initiators. If these contaminants are more or less harmless, oxidation proceeds rather slowly. However, in the presence of metallic salts, the process goes quite quickly, and the tablets may deteriorate even within their theoretical shelf-life.

The second aspect is the oxygen availability in tablets and substances. It is evident that the tablets have a low concentration of O_2_, so the reactions occur under an oxygen shortage. On the contrary, there is an excess of O_2_ in the substance, so oxygen does not limit the reaction rate. These reasons explain why the rate constants in the tablet differ from those in the substances. There are even more subtle matters involved in the reaction mechanism, but we will not touch on them here.

The solution in which *k*_1_ has the same value, but *k*_2_ is different for the tablets and substances is rather unusual and it should be explained. The constant *k*_1_ means that the presence of O_2_ does not matter in the first step, so we must conclude that impurities are the main players here. It is their concentration that regulates the rate of the reaction. The tablets are produced from the substances (API and excipients), so it is reasonable to assume that concentrations of contaminants in the tablets and substances are the same. This could explain the fact that *k*_1_ does not change.

The second finding that *k*_2_ is different is more difficult to explain. We can only suppose that the oxidation mechanism in the second stage is not the same as in the first. We have assumed that in the first stage a degenerate chain reaction occurs with the participation of impurities acting as initiators. Due to the hydroxyl group (clearly seen near 5000 cm^−1^ in the NIR spectra), *Inter* is itself rather reactive with oxygen without an initiator. Thus, its transformation into *Prod* (stage 2) is controlled by the O_2_ concentration, which is much higher in the substance than in the tablet. This possible mechanism for the second stage could explain why *k*_2_ in the second stage is 2.5 times greater than in the first.

We understand that moving from one stage of data processing to the other, we extract more information regarding the intermediate and final products, but on the other hand, we inevitably add extra noise to spectral data. This is the reason why we do not analyze the final ‘pure’ spectra of *Inter* and *Prod* in detail. At the same time, using multi-step MCR and kinetic modeling, we managed to propose the mechanism of degradation of a rather complex object, sealed Diclofenac tablets.

## Figures and Tables

**Figure 1 molecules-27-07904-f001:**
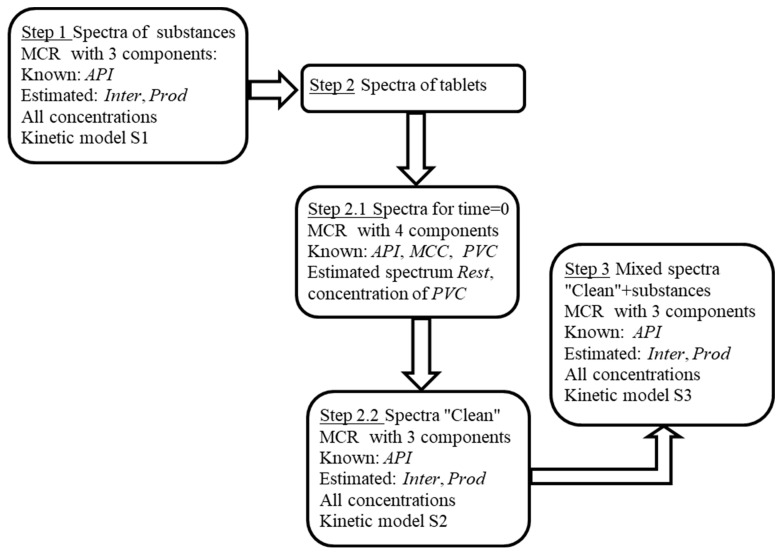
Scheme of multi-step MCR + Kinetics analysis.

**Figure 2 molecules-27-07904-f002:**
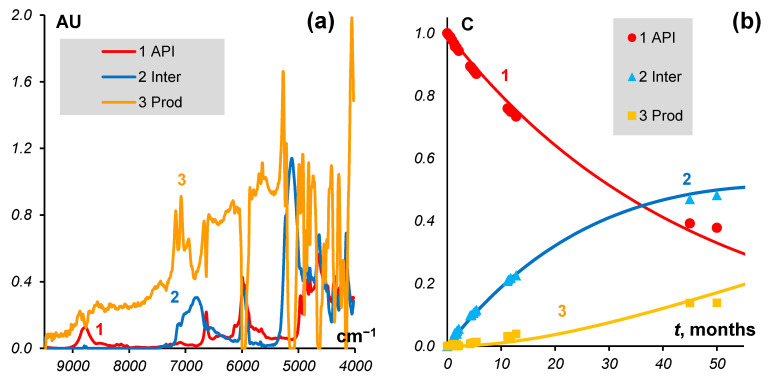
MCR model for substances. (**a**) Spectra of *API*, *Inter*, and *Prod* (**b**) Concentrations (markers) and kinetics (curves).

**Figure 3 molecules-27-07904-f003:**
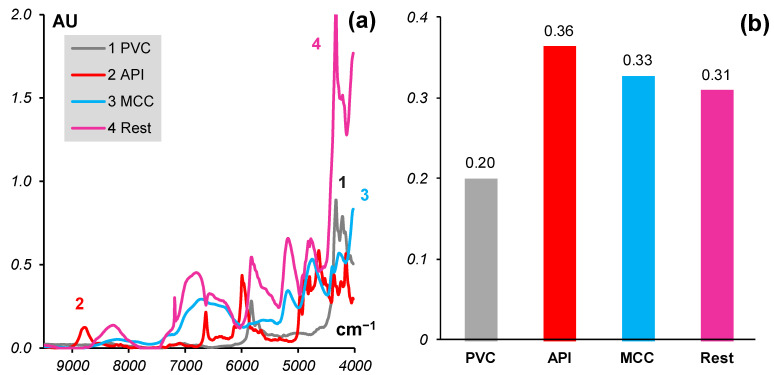
(**a**) Spectrum of *Rest* and spectra of other components; (**b**) Known (*API*, *MCC*, *Rest*) and estimated (*PVC*) concentrations of components.

**Figure 4 molecules-27-07904-f004:**
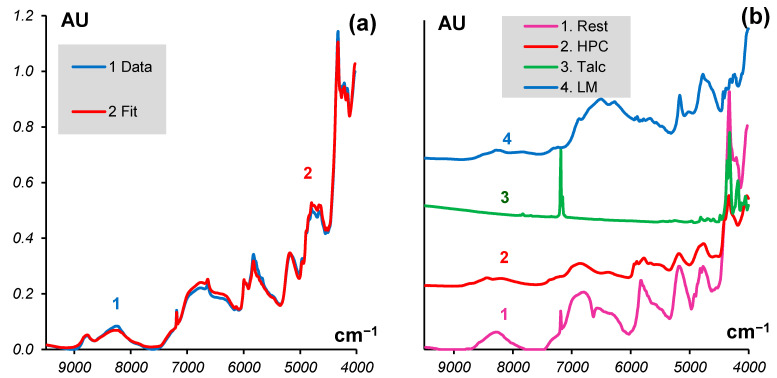
(**a**) Experimental spectrum (1) and its fit by MCR (2); (**b**) Spectrum of *Rest* and the spectra of some common excipients (vertically shifted).

**Figure 5 molecules-27-07904-f005:**
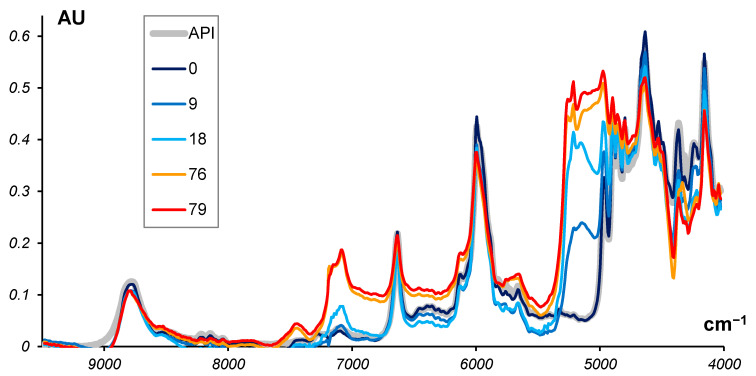
The ‘clean’ spectra of tablets after natural aging at five time points: 0, 9, 18, 76, and 79 months. The spectrum of API is shown as a reference.

**Figure 6 molecules-27-07904-f006:**
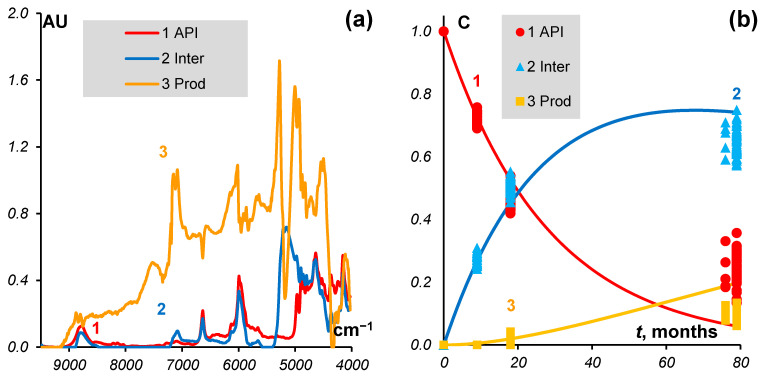
MCR model for tablets. (**a**) Spectra of *API*, *Inter,* and *Prod*; (**b**) Concentrations (markers) and kinetics (curves).

**Figure 7 molecules-27-07904-f007:**
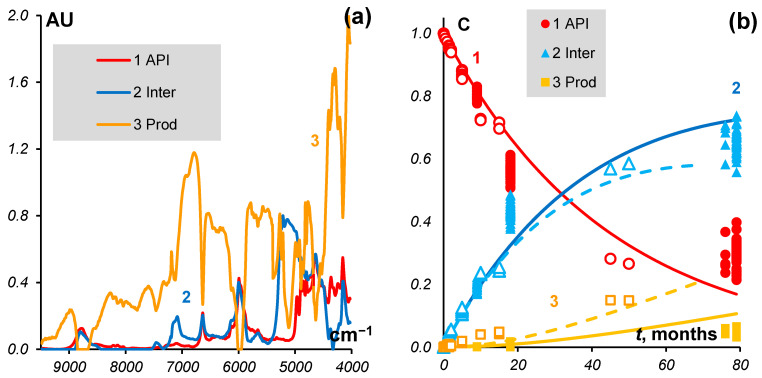
MCR model for the combined data. (**a**) Spectra of *API*, *Inter*, and *Prod*. (**b**) Concentrations (markers) and kinetics (curves). Solid curves and closed markers are for tablets, dashed curves and open markers are for substances.

**Table 1 molecules-27-07904-t001:** Tablet components.

Component Code	Role	Concentration	Spectrum
*PVC*	Blister	See text	Known
*MCC*	Excipient	Known	Known
*Rest*	Other excipients and coating	Known	Unknown
*API*	Diclofenac sodium	Known	Known
*Inter*	Intermediate of API degradation	Unknown	Unknown
*Prod*	Product of API degradation	Unknown	Unknown

**Table 2 molecules-27-07904-t002:** The kinetic constants.

*k* _1_	0.022	both
*k* _21_	0.003	tablets
*k* _22_	0.007	substances

## Data Availability

Not applicable.

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
