# Peer review of "Kinetic Model of Diclofenac Degradation Developed Using Multivariate Curve Resolution Method"

_molecules, 2022, doi:10.3390/molecules27227904_

Round 1

Reviewer 1 Report

The work is very interesting, especially for cognitive reasons, but practical applications can be expected. Regarding the results and discussions, I have generally no comments, but the introduction needs to be revised in order to consider the publication of the manuscript.
The first introduction begins with the sentence "Diclofenac is a popular, widely used, nonsteroidal anti-inflammatory drug" - this is of course true, the drug is even available as OTC. Unfortunately, the cited work in [1] comes from 1988. In the introductory part of the introduction, please discuss the mechanism of action, take into account the processes of generating interactions (also with ASA in patients treated for coronary artery disease). I believe that it is necessary to introduce the reader to issues that may arise from the consequences of the drug form degradation process.
2.Many aspects in the introduction and discussion are based on old works. The exceptions are self-citations (items 5, 6, 7 - but I would like to point out that they are justified by the subject) and the reference to the European drug characteristics. Please refer to current literature.
3. In summary, please present the most important aspects of the work also for practical reasons.

Author Response

We thank the Editor and  Reviewers for their comments and remarks, which helped us in improving readability of our manuscript. All corresponding changes are highlighted in the MS in ‘Track changes’ mode.

Minor language fixes made.

Point by point answers are presented below, reviewers’ comments are in Italics, our answers are in regular font.

Comments and Suggestions for Authors

The work is very interesting, especially for cognitive reasons, but practical applications can be expected. Regarding the results and discussions, I have generally no comments, but the introduction needs to be revised in order to consider the publication of the manuscript.

The first introduction begins with the sentence "Diclofenac is a popular, widely used, nonsteroidal anti-inflammatory drug" - this is of course true, the drug is even available as OTC. Unfortunately, the cited work in [1] comes from 1988. In the introductory part of the introduction, please discuss the mechanism of action, take into account the processes of generating interactions (also with ASA in patients treated for coronary artery disease). I believe that it is necessary to introduce the reader to issues that may arise from the consequences of the drug form degradation process.

Additional information about Diclofenac is added in Introduction.

  1. Many aspects in the introduction and discussion are based on old works. The exceptions are self-citations (items 5, 6, 7 - but I would like to point out that they are justified by the subject) and the reference to the European drug characteristics. Please refer to current literature

We decided to leave Ref [1] as relevant , but changed others.  References  2-4 have been removed and 5 new have been added(ref. 2-ref.6).

  1. In summary, please present the most important aspects of the work also for practical reasons.

The following text is added into abstract:

We study tablets in blister packs, exactly as they are presented in pharmacies, and this is important from a practical point of view

Reviewer 2 Report

This paper presents the application of the MCR-ALS and kinetic modelling of the results to describe the degradation of diclofenac. The study is an interesting alternative to standard methods applied to test the degradation of APIs.

My comments are as follows:

The abstract presents the overall idea of the study and the performed analyses, but it should also shortly summarize the main result.

Please support the statements in lines 25-29 with literature (ex. ‘there are cases where experts have observed…’).

Lines 52-61: The authors provided a very good explanation of their study. However, in my opinion the justification of the adapted approach should follow the description of the applied methods. I would transfer this part to the following sections of the paper, so that the methodological approach is simply described first. In my opinion, some sentences from the introduction would better match the Discussion. Please consider transferring all the explanation and justification phrases to the Discussion.

I think it would be helpful to add short explanation of the general principles of the MCR-ALS and what do “soft”, “grey” and “hard” approaches mean, even though suitable references are provided.

Lines 72-77: The Authors claim that using this analysis, only a global kinetic model and approximated results can be obtained. There exists a considerable body of literature on the degradation of diclofenac – what are the advantages and the novelty of this study then?

Although the need to conduct the study is presented, the objectives of the present paper should be stated clearly at the end of the Introduction too.

Why did the Authors assume that there would be only one intermediate and one degradation product of diclofenac? In course of the degradation of one API several intermediates and products may appear.

Table 1.: if the concentration of some components is known, is it possible to include these values in the table?

Figure 1, Step 1: There is a typo in the word “concentrations”, please correct.

The results are presented clearly and in a simple way.

Discussion: the role of the oxygen is first mentioned in the discussion. In my opinion, general information on oxidation processes should be given in the Introduction. The discussion should be supported with sources.

Moreover, lines 269-271: “The tablets are produced from the substances (API and excipients), so it is reasonable to assume that concentrations of contaminants in the tablets and substances are the same.”. In my opinion this is a very simplified assumption, as API is not the only component of the tablet that could be contaminated – excipients could be contaminated too.

Author Response

Comments and Suggestions for Authors

We thank the Editor and  Reviewers for their comments and remarks, which helped us in improving readability of our manuscript. All corresponding changes are highlighted in the MS in ‘Track changes’ mode.

Minor language fixes made.

Point by point answers are presented below, reviewers’ comments are in Italic, our answers are in regular font.

Comments and Suggestions for Authors

This paper presents the application of the MCR-ALS and kinetic modelling of the results to describe the degradation of diclofenac. The study is an interesting alternative to standard methods applied to test the degradation of APIs.

My comments are as follows:

The abstract presents the overall idea of the study and the performed analyses, but it should also shortly summarize the main result.

The following text is added to abstract.

... helps to propose a generic degradation mechanism, which includes: a global kinetic model; approximations of the NIR spectra of the intermediate and the product; rough estimates of rate constants

Please support the statements in lines 25-29 with literature (ex. ‘there are cases where experts have observed…’).

We wrote that these cases were revealed by ––

“...experts have observed a rapid degradation of this drug during routine monitoring or using complex laboratory tests. Every year since 2017, during routine testing, from one to three substandard batches of Diclofenac tablets have been identified.”

More details are explained below  ––

“The study was undertaken when unexpectedly rapid changes in the NIR spectra of the intact tablets were noticed in the course of routine monitoring of medicines following the workflow described in [8].”Lines 52-61: The authors provided a very good explanation of their study. However, in my opinion the justification of the adapted approach should follow the description of the applied methods. I would transfer this part to the following sections of the paper, so that the methodological approach is simply described first. In my opinion, some sentences from the introduction would better match the Discussion. Please consider transferring all the explanation and justification phrases to the Discussion.

Agree. We transferred this to the discussion section.

I think it would be helpful to add short explanation of the general principles of the MCR-ALS and what do “soft”, “grey” and “hard” approaches mean, even though suitable references are provided.

We tried not to overload the paper with unnecessary details. The explanation of the soft MCR-ALS, which is used in the paper, is presented in section 2.3. As for other versions (hard and grey), they are not used here, so we consider their descriptions redundant. The interested readers can follow the presented references.

Lines 72-77: The Authors claim that using this analysis, only a global kinetic model and approximated results can be obtained. There exists a considerable body of literature on the degradation of diclofenac – what are the advantages and the novelty of this study then?

We agree with this comment and have added a short comment in the introduction ––

There are many works on the degradation of Diclofenac. However, they are devoted to cases of degradation in the environment (wastewater); see, for example [5, 6]. To the best of our knowledge, studies of aging of the drug in packaged tablets have not yet been conducted.

The advantages and novelty of our study are explained as follows

“The principal advantage of our study is that we use data that reflects the natural aging of Diclofenac over a rather long period, up to (or close to) its complete degradation. Another advantage is the inclusion of the substance-only data, which adds new information, but at the same time produces new problems in the data interpretation. “ (page 2)

Although the need to conduct the study is presented, the objectives of the present paper should be stated clearly at the end of the Introduction too.

They are here (page 3) ––

“We are going to present: a global kinetic model; approximations of the NIR spectra of the intermediate and the product; rough estimates of rate constants.”

Why did the Authors assume that there would be only one intermediate and one degradation product of diclofenac? In course of the degradation of one API several intermediates and products may appear.

Any modeling is carried out within the framework of certain assumptions, which are determined not only by our understanding of the ongoing processes, but also by the available experimental data. Of course, there may be more intermediates and products involved in the degradation process, but the data we have do not allow us to go further. Thus we have to present the global (gross) kinetic model with some generalized components.  Please note that we claimed ––

“we will not be able to demonstrate: a detailed kinetic mechanism; identification of intermediates and products; accurate estimates of the rate constants.” (lines 75-78)

Table 1.: if the concentration of some components is known, is it possible to include these values in the table?

The purpose of the table is to give an idea of what is known and unknown in the tablet, so the concentration values here are redundant. The composition of the tablets is described at the beginning of section 2.1 in the form used in the pharmacy.

Figure 1, Step 1: There is a typo in the word “concentrations”, please correct.

Thank you, corrected.

The results are presented clearly and in a simple way.

Discussion: the role of the oxygen is first mentioned in the discussion. In my opinion, general information on oxidation processes should be given in the Introduction. The discussion should be supported with sources.

Agree. This part is transferred to introduction.

Moreover, lines 269-271: “The tablets are produced from the substances (API and excipients), so it is reasonable to assume that concentrations of contaminants in the tablets and substances are the same.” In my opinion this is a very simplified assumption, as API is not the only component of the tablet that could be contaminated – excipients could be contaminated too.

Our assumption about the similarity of the mechanism of degradation of tablets and substances is based on the similarity of the results of NIR measurements of tablets and substances in their initial state and in the process of degradation. We are sure that friar William of Ockham would have supported us in this decision.

Round 2

Reviewer 1 Report

Manuscript was improved, and IMHO may be considered for publication in present form.